# Optimising 2D Pose Representation: Improving Accuracy, Stability and Generalisability in Unsupervised 2D-3D Human Pose Estimation

## Abstract

This paper addresses the problem of 2D pose representation during unsupervised 2D to 3D pose lifting to improve the accuracy, stability and generalisability of 3D human pose estimation (HPE) models. All unsupervised 2D-3D HPE approaches provide the entire 2D kinematic skeleton to a model during training. We argue that this is sub-optimal and disruptive as long-range correlations are induced between independent 2D key points and predicted 3D ordinates during training. To this end, we conduct the following study. With a maximum architecture capacity of 6 residual blocks, we evaluate the performance of 5 models which each represent a 2D pose differently during the adversarial unsupervised 2D-3D HPE process. Additionally, we show the correlations between 2D key points which are learned during the training process, highlighting the unintuitive correlations induced when an entire 2D pose is provided to a lifting model. Our results show that the most optimal representation of a 2D pose is that of two independent segments, the torso and legs, with no shared features between each lifting network. This approach decreased the average error by 20% on the Human3.6M dataset when compared to a model with a near identical parameter count trained on the entire 2D kinematic skeleton. Furthermore, due to the complex nature of adversarial learning, we show how this representation can also improve convergence during training allowing for an optimum result to be obtained more often. **Code and weights will be made available**

## 1 Introduction

Monocular 3D human pose estimation (HPE) aims to reconstruct the 3D skeleton of the human body from 2D images or video. This is known to be an ill-posed inverse problem as multiple different 2D poses can correspond to the same 3D pose. Even with this hurdle, deep learning has allowed for accurate 2D-3D pose regression mappings to be learned allowing for remarkable results when trained and tested on 3D pose datasets (Wandt & Rosenhahn, 2019; Martinez et al., 2017; Pavlakos et al., 2018; Yang et al., 2018; Cheng et al., 2020; Pavllo et al., 2019). Unfortunately, the difficulty of obtaining 3D datasets leads to poor performance when evaluating in domains where rigid environmental constraints (lighting, action, camera location, etc) are unable to be controlled. Recent work (Chen et al., 2019; Wandt et al., 2022; Drover et al., 2019; Yu et al., 2021) has investigated if an unsupervised solution for 3D HPE is possible. These approaches utilize a geometric self-supervision cycle through random rotations to create a consistent lifting network, with some form of pose or probability discriminator to see if the rotated pose, once reprojected back to 2D, is realistic. As 2D data is cheaper to obtain, more efficient for computations and readily available in many circumstances, improving the performance of unsupervised 2D-3D HPE networks would therefore allow for accurate 3D poses to be obtained in many unconstrained scenarios.

An overlooked aspect in prior work however, is the representation of the 2D pose being given to the lifting model. We posit that when a full 2D pose is provided to a lifting model during training, long-range correlations are induced between a key points 3D prediction and all of the poses' other 2D key point coordinates (i.e. the 3D prediction of the left elbow will be influenced by the 2D coordinate of the right ankle). Although supervised approaches have touched upon this topic by learning joint dependency through graph convolutional networks (GCN) (Lee et al., 2018; Zhao

et al., 2019), relationships between joints via relational networks Park & Kwak (2018) or splitting and recombining the limbs of a pose (Zeng et al., 2020). To our best knowledge, this has never been done in an unsupervised setting, and it has never been assumed that the pose could be two or more independent structures with no inter-relational correspondence needed. Additionally, the large variations in network architecture and optimisation within prior work mean we are unable to fairly compare the results between approaches to find an optimum representation.

We address this problem by training 5 models with near identical amounts of parameters and identical training approaches on different 2D pose representations. By evaluating the results obtained from each model we will be able to determine an optimum representation that future work can use to obtain the best performance. We also show the correlations induced between 2D key points during training when a full pose is provided to a model as well as our best performing 2D representation model, providing some intuition behind the improved performance. To summarise our paper makes the following contributions:

- We show the effect when using different 2D pose representations for the unsupervised adversarial 2D-3D HPE process, where by changing the 2D pose representation we can reduce the average error by 20%.

- Our findings can be easily implemented within current the state of the art as our approach utilizes the popular residual block introduced by Martinez (Martinez et al., 2017) and used within (Chen et al., 2019; Wandt et al., 2022; Wandt & Rosenhahn, 2019; Drover et al., 2019; Yu et al., 2021).

- We show the correlations induced between key points for a full 2D pose representation model and our best 2D pose representation model highlighting the sub-optimal learning when a full 2D pose is provided to a network.

- We show the adversarial stability of our best pose representation model against a full 2D pose representation model, highlighting that the improvement is consistent across multiple random initializations.

## 2 RELATED WORK

There currently exists two main avenues of deep-learning research for 3D HPE. The first learns the mapping of 3D joints directly from a 2D image (Pavlakos et al., 2017; Lie et al., 2019; Mehta et al., 2017a; Li et al., 2015; Tome et al., 2017). The second builds upon an accurate intermediate 2D pose estimate, with the 2D pose obtained from an image through techniques such as Stacked-Hourglass Architectures (Newell et al., 2016) or Part Affinity Fields (Cao et al., 2021), and lifts this 2D pose to 3D. This work focuses on the latter 2D to 3D lifting avenue which can be organized into the following categories:

### 2.1 FULLY SUPERVISED

Fully supervised approaches seek to learn mappings from paired 2D-3D data which contain ground truth 2D locations of key points and their corresponding 3D coordinates. Martinez et al. (2017) introduced a baseline fully connected regression model which learned 3D coordinates from their relative 2D locations. Exemplar approaches such as Chen & Ramanan (2017) and Yang et al. (2019) use large dictionaries/databases of 3D poses with a nearest-neighbour search to determine an optimal 3D pose. Pavllo et al. (2019) used temporal convolutions over 2D key points to predict the pose of the central or end frame in a time series, whereas Mehta et al. (2017b) utilized multi-task learning to combine a convolutional pose regressor with kinematic skeleton fitting for real-time 3D HPE.

### 2.2 WEAKLY-SUPERVISED

Weakly-Supervised approaches do not use explicit 2D-3D correspondences and instead use either augmented 3D data during training or unpaired 2D-3D data to learn human body priors (shape or articulation). Pavlakos et al. (2018) and Ronchi et al. (2018) proposed the learning of 3D poses from 2D with ordinal depth relationships between key points (e.g. the right wrist is behind the right elbow). Wandt & Rosenhahn (2019) introduced a weakly-supervised adversarial approach where

they transformed their predicted and ground truth 3D poses into a kinematic chain (Wandt et al., 2018) before being seen by a Wasserstein critic network (Gulrajani et al., 2017). Yang et al. (2018) lifted 2D poses from in the wild scenes where no ground truth data was available with a critic network that compared these predictions against ground-truth 3D skeletons. By contrast, Zhou et al. (2017) did not use a critic network but transfer learning that used mixed 2D and 3D labels in a unified network to predict 3D poses in scenarios where no 3D data is available. Drover et al. (2019) investigated if 3D poses can be learned through 2D self-consistency alone, where they would rotate a predicted 3D pose and reproject it back into 2D before passing it back to the model for comparison. This led to the discovery that a 2D critic network was also needed and self-consistency alone is not sufficient.

## 2.3 UNSUPERVISED

Unsupervised approaches do not utilize any 3D data during training, unpaired or otherwise. Kudo et al. (2018) introduced one of the first unsupervised adversarial networks which utilized random reprojections and a 2D critic network, under the assumption that any predicted 3D pose once rotated and reprojected should still produce a believable 2D pose. Chen et al. (2019) expanded this work and that of Drover et al. (2019) by introducing an unsupervised adversarial approach with a self-consistency cycle. They also provided ablation studies highlighting a 7% improvement when using temporal cues during training. Yu et al. (2021) built upon Chen et al. (2019) highlighting that these temporal constraints may hinder a models performance due to balancing multiple training objectives simultaneously and proposed splitting the problem into both a lifting and scale estimation module.

## 2.4 PRIOR WORK ON 2D POSE REPRESENTATIONS

Park & Kwak (2018) introduced one of the first studies that investigated if the human pose could be constructed of multiple interrelated structures. By utilizing relational networks, which were passed limb pairs during training ([left arm and right arm], [left arm and right leg], etc.), they aimed to discover shared feature information between limbs that would aid in the 2D-3D lifting process. Lee et al. (2018) utilized a propagating LSTM architecture where each LSTM would sequentially add another key point to its input. This intrinsic approach allowed joint relationships to be learnt0 over time as each key point was added. Jiang (2010) worked on an exemplar approach where their 3D dictionary was split into torso and legs key points during lookup. Though their idea behind this was to speed up the nearest-neighbour search process, they may have improved the generalisability of their model as it could find similar legs in one training example and similar torsos within another, increasing the number of possible library combinations. Zeng et al. (2020) investigated splitting a pose into different parts to learn localized correlations between key points first then global correlations between these localized segments afterward. Unlike our study however, Zeng et al. (2020) is a supervised network that still assumes that an entire 2D pose is one independent structure via feature sharing between localized groups throughout their network. Our study utilizes completely unsupervised networks, where two of the pose representations we investigate are independent with the idea that localized key-point groups could be unrelated to one another.

## 3 METHODOLOGY

In this section we describe the different 2D representations used and the intuition behind each representation. We then explain the adversarial learning approach of our various models.

## 3.1 2D POSE REPRESENTATIONS

Our 2D poses consist of $N$ key points $(x_i, y_i)$, $i = 1...N$, with the root key point, the midpoint between the left and right hip joint, being located at the origin $(0, 0)$. Because of the fundamental scale ambiguity associated with monocular images, it is impossible to infer absolute depth from a single view alone Nishimura et al. (2020). Therefore, we used max-normalisation on each of our 2D poses to scale their 2D key point coordinates between -1 and 1. In our study we investigate the effects of using the following five 2D pose representations for the unsupervised 2D-3D HPE process:

- **Full 2D Pose Representation:** The standard approach where a model is taught to predict the 3D ordinates from the entire 2D pose. The theory being that a good model should learn to correlate co-dependent key points and disassociate independent ones. In this scenario, we will have one network take an entire 2D pose as input and predict the 3D ordinates for the full pose.

- **Split and Recombine Leg and Torso key points:** A concept introduced by Zeng et al. (2020), the idea behind this representation is to first learn localized key point correlations, followed by the global context of a pose. For this representation, two separate networks will first learn the localized correlations between the leg and torso key points and output a vector of features learned. A further network will learn to combine these features and predict the 3D ordinates for the full pose.

- **Independent Leg and Torso key points:** In many actions (e.g. waving, eating, sitting etc) the legs and torso of a person can be more independent than co-dependent. Therefore another 2D pose representation will be that of completely independent torso and leg key points. For this representation we will have two independent lifting networks, one that accepts the 2D key points and predicts the 3D ordinates for the torso, and the other the 2D key points and 3D ordinates of the legs.

- **Split and Recombine 5 Limbs key points:** As humans, we are able to move each limb independently of one another. For this representation, 5 separate networks will learn the localized correlations of each limbs (left arm, right arm, torso, right leg and left leg) key points and output a vector of features learned. A further network will learn to combine these features and predict the 3D ordinates for the full pose.

- **Independent 5 Limbs key points:** Similar to the Independent leg and torso representation, this approach will look at the possibility of treating the entire pose as 5 independent limbs. For this representation, 5 separate networks will each take as input and predict the 3D ordinates for each limb independently (i.e. one network will take 3 key points as input belonging to the right arm and predict the 3D ordinates for the right arm).

## 3.2 LIFTING NETWORK

The lifting networks $(G)$ used are fully connected neural networks whose architecture was based on Martinez et al. (2017) and can be seen in Figure 3 in Appendix A. These predict one 3D ordinate relative to the root for each 2D key point provided:

$$G(\mathbf{x}, \mathbf{y}; \mathbf{w}) = \hat{\mathbf{z}} \tag{1}$$

where $(\mathbf{x}, \mathbf{y})$ are the 2D coordinates provided to the network, $\hat{\mathbf{z}}$ are the predicted 3D ordinate for each key point and $\mathbf{w}$ are the weights of our model learned during training. Once our lifting networks had made their predictions they are concatenated with the original 2D key points to create our final predicted 3D pose $(\mathbf{x}, \mathbf{y}, \hat{\mathbf{z}})$. A more detailed architecture explanation of each pose representation model can be seen in Appendix A.

## 3.3 REPROJECTION CONSISTENCY

Similar to prior work (Chen et al., 2019; Drover et al., 2019; Yu et al., 2021), we utilize a self-consistency cycle through random 3D rotations to reproject our predicted 3D poses to new synthetic 2D viewpoints. Let $\mathbf{Y} \in \mathbb{R}^{N \times 2}$ be our full 2D pose. Once a prediction $G(\mathbf{Y})$ is made and a full 3D pose $(\mathbf{x}, \mathbf{y}, \hat{\mathbf{z}})$ obtained, a random rotation matrix $\mathbf{R}$ will be created by uniformly sampling an azimuth angle between $[-\frac{8\pi}{9}, \frac{8\pi}{9}]$ and an elevation angle between $[\frac{-\pi}{18}, \frac{\pi}{18}]$. The predicted 3D pose will be rotated by this matrix and reprojected back to 2D via projection $\mathbf{P}$, obtaining a new synthetic viewpoint of the pose and the matrix $\tilde{\mathbf{Y}} \in \mathbb{R}^{N \times 2}$ where $\tilde{\mathbf{Y}} = \mathbf{PR}[(\mathbf{x}, \mathbf{y}, \hat{\mathbf{z}})]$. To enable our model to learn consistency, if we now pass $\tilde{\mathbf{Y}}$ to the same lifting network, perform the inverse rotation $\mathbf{R}^{-1}$ on the newly predicted 3D pose $(\tilde{\mathbf{x}}, \tilde{\mathbf{y}}, \tilde{\hat{\mathbf{z}}})$ and reproject it back into 2D, we should obtain our original matrix of 2D key points $\mathbf{Y}$. This cycle allows our lifting networks to learn self-consistency during training where they seek to minimize the following components in the loss function:

$$\mathcal{L}_{2D} = \|\mathbf{Y} - \mathbf{PR}^{-1}[G(\tilde{\mathbf{Y}})]\|^2 \tag{2}$$

In the case of our independent leg and torso and independent 5 limb 2D pose approaches, each network will receive its own $\mathcal{L}_{2D}$ loss based on the error between the key points that they predicted. As an example, part of the $\mathcal{L}_{2D}$ loss for the right arm lifter within our independent 5 limbs representation would include the difference between the original 2D key point coordinate of the right wrist, and its 2D coordinate once $\tilde{\mathbf{Y}}$ was inversely rotated and reprojected. This error is not included in the $\mathcal{L}_{2D}$ loss for the left arm lifter as it does not predict the 3D ordinate for this key point.

### 3.4 90 DEGREE CONSISTENCY

During our study, we found that increasing self-consistency was key to reducing the evaluation error (see Appendix B). Therefore, we introduce new self-consistency constraints during training based on rotations around the $y$ axis at $90°$ increments. Let $(\mathbf{x}, \mathbf{y}, \hat{\mathbf{z}})$ be the predicted 3D pose from our model. If we assume a fixed camera position and rotate our pose $90°$, then the depth component of our pose $(\hat{\mathbf{z}})$ prior to rotation will now lie on the $x$ axis from our cameras viewpoint. As our poses are normalised between -1 and 1, a $90°$ clockwise rotation of the 3D pose $(\mathbf{x}, \mathbf{y}, \hat{\mathbf{z}})$ will produce the pose $(\hat{\mathbf{z}}, \mathbf{y}, -\mathbf{x})$. Therefore, providing $(\hat{\mathbf{z}}, \mathbf{y})$ as input to our generators should result in $-\mathbf{x}$ as its predictions. This fact allows for the inclusion of three additional consistency constraints in the loss function of our generators which teach consistency at a $90°$ clockwise rotation, a $90°$ anticlockwise rotation and a $180°$ rotation, which are as follows:

$$\|G(\hat{\mathbf{z}}, \mathbf{y}; \mathbf{w}) + \mathbf{x}\|^2 = 0 \tag{3}$$

$$\|G(-\hat{\mathbf{z}}, \mathbf{y}; \mathbf{w}) - \mathbf{x}\|^2 = 0 \tag{4}$$

$$\|G(\mathbf{x}, \mathbf{y}; \mathbf{w}) + G(-\mathbf{x}, \mathbf{y}; \mathbf{w})\|^2 = 0 \tag{5}$$

The left parts of these constraints are summed in the final loss function to produce $\mathcal{L}_{90°}$ and included in the optimisation function of our models. Similar to $L_{2D}$ in the case of our independent 2D pose representation networks, each lifting network will have its own $L_{90°}$ depending on the 2D key points that it predicted for. Although we could include three similar constraints for $90°$ rotational increments around the $x$ axis, we found that these hinder the performance of the model. This is due to $90°$ $x$ axis rotations producing a birds eye and ground-up view of a 2D pose, which contains little variation between 2D key points.

### 3.5 DISCRIMINATOR LOSS

Although self-consistency is important, alone it is insufficient for generating realistic 3D skeletons (Chen et al., 2019). Therefore, we utilize a 2D discriminator $D$, that takes as input a 2D pose and outputs a probability value of that pose being plausible or unrealistic. The architecture of our discriminator is a fully connected neural network made of 3 residual blocks (Martinez et al., 2017) and ending with a linear prediction function. It learns to discriminate between the real 2D poses within our data $\mathbf{Y}$, and our reprojected 2D pose $\tilde{\mathbf{Y}}$. This provides feedback to the generators during training, enabling the learning of geometric priors such as joint angles and limb length ratios. Our discriminator $D$ utilized the least squares loss (Mao et al., 2017) as we found it performed better than the standard adversarial loss during our experiments:

$$\begin{aligned} \min_D &= \frac{1}{2}\mathbb{E}[D(\mathbf{Y}) - 1]^2 + \frac{1}{2}\mathbb{E}[D(\tilde{\mathbf{Y}})]^2 \\ \min_G &= \frac{1}{2}\mathbb{E}[D(\tilde{\mathbf{Y}}) - 1]^2 \end{aligned} \tag{6}$$

Unlike our consistency constraints, we do not provide a unique version of $L_{adv}$ to each lifting network in our independent 2D pose representation networks and instead provide the same loss (with a different weight) to each network. This is due to two reasons; Firstly, although our independent lifting networks are being trained separately, we still want them to produce a believable reprojected 2D pose together and having one discriminator see the entire pose will provide this feedback during training. Secondly, we found that trying to discriminate between segments of a 2D pose provided poor feedback during training. An example being that 2D legs and arms are normally represented as straight or bent lines, making it hard to discriminate between plausible and implausible.

## 3.6 TRAINING

As discussed, our lifting networks were trained adversarially with a random reprojection and $90°$ consistency constraints. The network parameters are then updated to optimise the total loss for each lifting model given by:

$$\mathcal{L} = w_1\mathcal{L}_{adv} + w_2\mathcal{L}_{2D} + w_3\mathcal{L}_{90°} \qquad (7)$$

where, $w_1$, $w_2$ and $w_3$ are the relative weights for the adversarial loss component, reprojection consistency loss component and $90°$ consistency loss component respectively and defined numerically in Appendix A. We trained our models unsupervised following (Chen et al., 2019) using a batch size of 8192 and the Adam optimiser (Kingma & Ba, 2015) with a learning rate of 0.0002. Our experiments use $N = 16$ key points. During discriminator training we employ label flipping with a 10% chance.

## 4 EVALUATION AND RESULTS

Here we compare and evaluate the performance of our different 2D pose representation models against each other and various state-of-the-art models. Though our aim is not to beat the current state of the art, we believe that comparing against current models is important to examine how just a simple change in 2D pose representation can compare against the improvement from more complex adjustments. Our results show that not only can an optimum 2D pose representation improve upon the default full pose representation by 20%, but it can also improve generalisability to unseen poses, even when compared against supervised and weakly-supervised approaches.

### 4.1 QUANTITATIVE RESULTS ON HUMAN3.6M

Human3.6M (H36M) (Ionescu et al., 2014) is one of the largest and most widely used 3D human pose datasets, containing 3.6 million 3D human poses. It consists of both video and motion capture (MoCap) data from 4 viewpoints of 5 female and 6 male subjects performing specific actions (e.g. talking on the phone, taking a photo, eating, etc.). There are two main evaluation protocols for the H36M dataset, which use subjects 1, 5, 6, 7 and 8 for training and subject 9 and 11 for evaluation. Both protocols report the Mean Per Joint Position Error (MPJPE), which is the Euclidean distance in millimeters between the predicted and ground truth 3D coordinates. We report the protocol-II performance of our model on all frames of the validation set which employs rigid alignment between the ground truth and predicted pose before evaluation. The results of our different 2D pose representation models can be seen in Table 1. From our results, we can see that representing a 2D pose as an independent torso and legs improves performance the most, as this representation achieved a 20% lower average error than our full 2D pose network. Additionally, our split-combine leg and torso and split-combine 5 limbs representation also showed improved performance, decreasing the average error by 6% and 9% respectively. Our independent 5 limbs model did not improve results which may be due to this representations models using 3 or 4 key points to predict 3D ordinates which may not contain enough information to learn the task at hand. By looking at our leg and torso results we can see it additionally performed well against several fully supervised models. In fact, by using this representation we achieved the second highest performance in the Discussing, Posing, Smoking and Walking action. Although improving the Discussing, Posing and Smoking action seems intuitive, as we would assume the arms and legs of a person to be independent during these actions, it is surprising that we achieved such high results in walking as we would expect to see larger levels of co-dependence between the torso and legs during this movement.

### 4.2 QUANTITATIVE RESULTS ON MPI-INF-3DHP

MPI-INF-3DHP (Mehta et al., 2017a) is a markerless MoCap dataset containing the 3D human poses of 8 actors performing 8 different activities. We use this dataset to highlight the effect that 2D pose representations have on a model's generalisability to unseen poses when trained on the H36M dataset. The evaluation metrics used are the *percentage of correctly positioned key points* (PCK3D) and *area under the curve* (AUC) as defined in (Mehta et al., 2017a). As our poses are normalised, we upscale our 3D poses by their original 2D normalizing factor before evaluation. Our results can be seen in Table 2. Similar to our H36M results, all of our 2D representation approaches, aside from our independent 5 limbs representation, performed better on unseen poses when compared to a full

Table 1: The reconstruction error (MPJPE) on the H36M validation set. **Legend**: (+) denotes extra data during training. (GT) denotes providing 2D ground truth key points to a network for prediction. (T) denotes the use of temporal information, all compairson results are taking from their respective papers, lower is better, best in bold, second best underlined.

| Method | Approach | Direct. | Discuss | Eat | Greet | Phone | Photo | Posing | Purchase |
|---|---|---|---|---|---|---|---|---|---|
| Martinez et al. (2017) (GT) | Supervised | 39.5 | 43.2 | 46.4 | 47.0 | 51.0 | 56.0 | 41.4 | 40.6 |
| Pavllo et al. (2019) (GT) | Supervised | 36.0 | 38.7 | 38.0 | 41.7 | 40.1 | 45.9 | 37.1 | 35.4 |
| Cai et al. (2019) (GT) | Supervised | 36.8 | 38.7 | 38.2 | 41.7 | 40.7 | 46.8 | 37.9 | 35.6 |
| Yang et al. (2018) (+) | Weakly-Supervised | 26.9 | 30.9 | 36.3 | 39.9 | 43.9 | 47.4 | 28.8 | 29.4 |
| Wandt & Rosenhahn (2019) (GT) | Weakly-Supervised | 33.6 | 38.8 | 32.6 | 37.5 | 36.0 | 44.1 | 37.8 | 34.9 |
| Pavlakos et al. (2018) (+) | Weakly-Supervised | 34.7 | 39.8 | 41.8 | 38.6 | 42.5 | 47.5 | 38.0 | 36.6 |
| Chen et al. (2019) (GT)(T) | Unsupervised | - | - | - | - | - | - | - | - |
| Yu et al. (2021) (GT)(T) | Unsupervised | - | - | - | - | - | - | - | - |
| Full 2D Pose Network (Ours)(GT) | Unsupervised | 51.5 | 47.1 | 48.1 | 53.7 | 47.6 | 56.2 | 45.9 | 46.1 |
| Split-Combine Leg and Torso Network (Ours)(GT) | Unsupervised | 45.3 | 44.8 | 43.4 | 47.2 | 46.2 | 48.2 | 43.2 | 41.5 |
| Independent Leg and Torso Network (Ours)(GT) | Unsupervised | 38.1 | 38.7 | 36.8 | 42.0 | 41.9 | 49.9 | 36.8 | 38.8 |
| Split-Combine 5 Limbs Network (Ours)(GT) | Unsupervised | 43.6 | 42.3 | 43.4 | 45.9 | 45.1 | 49.9 | 41.5 | 42.0 |
| Independent 5 Limbs Network (Ours)(GT) | Unsupervised | 90.0 | 98.1 | 109.4 | 94.5 | 100.3 | 109.3 | 74.0 | 105.5 |

| Method | Approach | Sit | SitD. | Smoke | Wait | Walk | WalkD. | WalkT. | Avg. |
|---|---|---|---|---|---|---|---|---|---|
| Martinez et al. (2017) (GT) | Supervised | 56.5 | 69.4 | 49.2 | 45.0 | 49.5 | 38.0 | 43.1 | 47.7 |
| Pavllo et al. (2019) (GT) | Supervised | 46.8 | 53.4 | 41.4 | 36.9 | 43.1 | 30.3 | 34.8 | 40.0 |
| Cai et al. (2019) (GT) | Supervised | 47.6 | 51.7 | 41.3 | 36.8 | 42.7 | 31.0 | 34.7 | 40.2 |
| Yang et al. (2018) (+) | Weakly-Supervised | 36.9 | 58.4 | 41.5 | 30.5 | 29.5 | 42.5 | 32.2 | 37.7 |
| Wandt & Rosenhahn (2019) (GT) | Weakly-Supervised | 39.2 | 52.0 | 37.5 | 39.8 | 34.1 | 40.3 | 34.9 | 38.2 |
| Pavlakos et al. (2018) (+) | Weakly-Supervised | 50.7 | 56.8 | 42.6 | 39.6 | 43.9 | 32.1 | 36.5 | 41.8 |
| Chen et al. (2019) (GT)(T) | Unsupervised | - | - | - | - | - | - | - | 51.0 |
| Yu et al. (2021) (GT)(T) | Unsupervised | - | - | - | - | - | - | - | 42.0 |
| Full 2D Pose Network (Ours)(GT) | Unsupervised | 55.4 | 63.9 | 47.0 | 47.5 | 46.5 | 50.7 | 46.6 | 50.0 |
| Split-Combine Leg and Torso Network (Ours)(GT) | Unsupervised | 55.3 | 66.9 | 45.2 | 43.5 | 39.8 | 47.4 | 43.1 | 46.8 |
| Independent Leg and Torso Network (Ours)(GT) | Unsupervised | 47.4 | 60.6 | 40.6 | 38.0 | 34.0 | 46.0 | 37.1 | 41.7 |
| Split-Combine 5 Limbs Network (Ours)(GT) | Unsupervised | 50.1 | 63.4 | 43.3 | 42.6 | 42.6 | 46.9 | 42.4 | 45.5 |
| Independent 5 Limbs Network (Ours)(GT) | Unsupervised | 123.1 | 154.6 | 96.6 | 86.1 | 87.4 | 108.4 | 96.0 | 102.2 |

Table 2: Results on the MPI-INF-3DHP dataset. Legend: (3DHP) denotes a model being trained on the MPI-INF-3DHP dataset. (H36M) denotes a model being trained on the Human3.6M dataset. (+) denotes additional training data. (*) uses transfer learning during from 2Dposenet. (T) denotes the use of temporal information during training, all comparison results are taking from their respective papers, higher is better, best in bold, second best underlined.

| Method | Approach | PCK3D | AUC |
|---|---|---|---|
| Mehta et al. (2017a) (3DHP)(H36M)(*) | Supervised | 76.5 | 40.8 |
| Zeng et al. (2020) (H36M) | Supervised | 77.6 | 43.8 |
| Yang et al. (2018) (H36M)(+) | Weakly-Supervised | 69.0 | 32.0 |
| Wandt & Rosenhahn (2019) (H36M) | Weakly-Supervised | 81.8 | 54.8 |
| Kanazawa et al. (2018) (3DHP)(T) | Weakly-Supervised | 77.1 | 40.7 |
| Chen et al. (2019) (3DHP)(T) | Unsupervised | 71.1 | 36.3 |
| Kundu et al. (2020) (H36M) | Unsupervised | 76.5 | 39.8 |
| Yu et al. (2021) (H36M)(T) | Unsupervised | 82.2 | 46.6 |
| Full 2D Pose Network (H36M)(Ours) | Unsupervised | 75.4 | 45.8 |
| Split-Combine Leg and Torso Network (H36M)(Ours) | Unsupervised | 76.1 | 47.5 |
| Independent Leg and Torso Network (H36M)(Ours) | Unsupervised | 78.5 | 48.5 |
| Split-Combine 5 Limbs Network (H36M)(Ours) | Unsupervised | 77.7 | 48.2 |
| Independent 5 Limbs Network (H36M)(Ours) | Unsupervised | 47.2 | 25.6 |

2D pose representation approach. Our independent leg and torso representation improved results the most with a 4% increase in PCK3D and 6% increase in AUC when compared to the full 2D pose network. Additionally, this representation achieved the second highest overall AUC metric, even compared to other supervised and weakly-supervised approaches.

## 5 REDUCING UNINTUITIVE KEY POINT CORRELATIONS

Our intuition behind the improved results when using an independent leg and torso representation when compared to the full 2D pose representation, is due to a model learning more intuitive correlations between 2D key points and 3D ordinates. To validate our assumption we performed the following additional study. By changing the 2D coordinate of a particular key point, we observed how the 3D predictions of our models were affected. From this, we could then see if any unintuitive correlations have been made between key points. For example, if we changed the key point location

for the left knee and the largest change in 3D prediction was for the right shoulder, then we can say that the model has learned to correlate the 2D key point of the left knee with its 3D prediction of the right shoulder. To conduct this study each 2D key point within the H36M validation set was scaled in 1% increments between -5% and 5% of its original size. Each key point was changed individually and we observed the change in prediction for each 3D ordinate caused by changing that particular key point. Figure 1 provides an example of the results from this study.

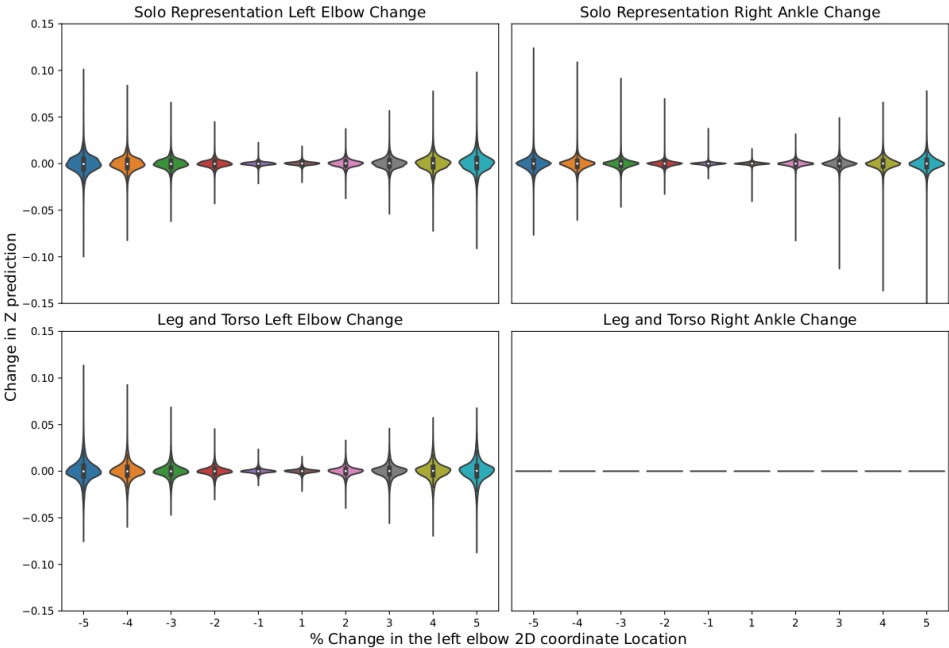

Figure 1: Showing the effect that changing the left elbows 2D key point coordinates has on the 3D ordinate prediction of the left elbow and right ankle for the full 2D pose representation model (top-left and top-right respectively), and the left elbow and right ankle for the independent leg and torso model (bottom-left and bottom-right respectively).

Examining Figure 1 we can see that the full 2D pose representation model has learned to correlate any change in the 2D coordinate for the left elbow with the 3D ordinate prediction of both the right ankle and left elbow. This highlights the unintuitive correlations learned when using a full 2D pose representation to train the model, as humans are able to move their left elbow and right ankle independently. By contrast, if we examine our independent leg and torso correlations, the correlation between the left elbow and right ankle has been removed while maintaining the correlation between the left elbows 2D coordinate and 3D ordinate prediction. We believe that by removing these unintuitive correlations we are able to obtain our improved results as our independent models predictions are less sensitive to a change within one specific key point. The full results of this study can be viewed within the supplementary material.

## 6   IMPROVING ADVERSARIAL TRAINING STABILITY

One fundamental question when utilising unsupervised networks is when to stop training. However, we find a lack of information within prior work detailing how long authors have trained their models for. Therefore, we assume that prior work trained for a set amount of epochs and picked the weights across these epochs that performed best on an validation set. Though fine from an evaluation viewpoint, in practice this would not work if we had no ground truth data. In a truly unsupervised scenario there are three approaches we could use to decide when to stop training. Firstly, we could monitor the discriminators loss and stop training when it too weak or strong. Though there is intuition for this approach, in practice a strong discriminator can cause a generator to fail due to vanishing gradients (Arjovsky & Bottou, 2017) and a weak discriminator provides poor feedback to a generator reducing its performance. Secondly, we could visualise the predictions per epoch and

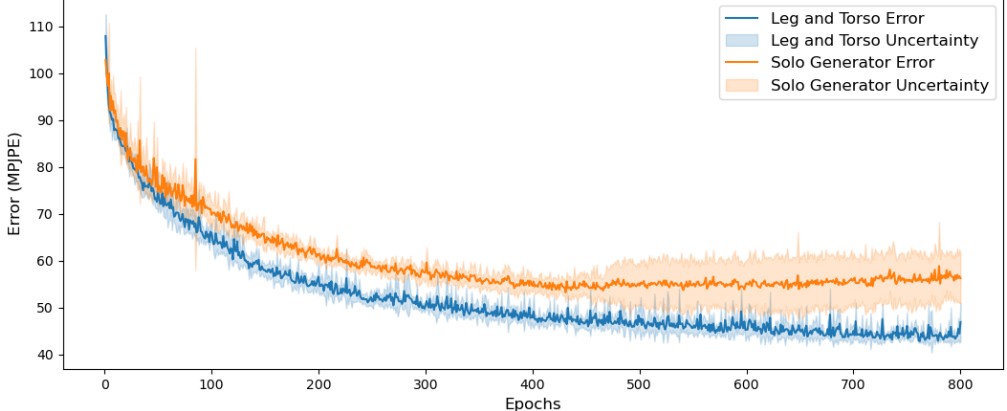

Figure 2: The average evaluation error (MPJPE) and uncertainty of 5 leg and torso generators compared against 5 solo generators on the H36M validation dataset for each training epoch.

decide by eye which pose is the best. Though having potentially hundreds of epochs and thousands of poses, this is not an efficient solution. Lastly, and most realistically, we could pick the final weight during the training of our model or average the weights between a certain range of epochs to use. For this scenario we show the stability of 5 solo and 5 independent leg and torso generators during adversarial training which can be seen in Figure 2 As we can see, by having a leg and torso generator training independently not only is the MPJPE lower, but all our models converge to a similar minima during adversarial training. This is especially apparent around epoch 450 where our solo generators' errors begin to diverge away from one another. However, our leg and torso generators' error does not exhibit this same pattern and the errors are tighter to the mean during training following a stable downwards curve. As these models were trained for 800 epochs, if we chose the last epochs weights to evaluate on then the average error of the leg and torso generators' would have been $47.0 \pm 4.0$mm and our solo generators' average error would have been $56.6 \pm 5.8$mm. From epoch 500 to 800 the average error and standard deviation of our leg and torso generators' was $45.6 \pm 2.0$mm, the average error and standard deviation of our solo generators' by comparison was $56.3 \pm 5.5$mm. Taking the epoch where we observed the minimum error for each model would give us a result of $42.61 \pm 0.57$mm for the leg and torso generators' and $51.55 \pm 2.13$mm for the solo generators'.

## 7    CONCLUSION

To conclude, we present a rigorous study investigating how different 2D pose representations affects the unsupervised adversarial 2D-3D lifting process. Our results show that the most optimum pose representation for this process is that of an independent leg and torso which reduced the average error by 20%. In fact, by having two lifting networks learn independently, we can improve both the accuracy and generalisability of a model. Furthermore, we have shown that when trained on a full pose, unintuitive correlations are induced between 2D coordinates and 3D ordinate predictions which we believe leads to poorer results. Additionally, we have shown that the best pose representation model can be recreated with greater consistency as different instances converge to similar minima during training whereas a full 2D pose representation approach diverges over time. Future work within this area could look at using attention mechanisms, which would allow the model to learn its own best representation for the 2D pose. We believe however, that for single frame scenarios it may fall victim to the same problems as our full 2D pose representation model, as our own experiments with selective attention have observed a model simply learning to get by with the initial 2D representation provided rather than change its own input. However, the main statement from this paper is that nearly all of our different 2D pose representations improved upon the standard full 2D pose representation, highlighting how a simple change to the input to your model can significantly improve your results.

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

## A    TRAINING APPROACH AND MODEL ARCHITECTURE FOR EACH 2D POSE REPRESENTATION

Here we will highlight the overall architectures and the optimisation functions with their numeric weights and reasoning for each of our 2D pose representation approaches. In Figure 3 you can see the residual block architecture which was included within all of our 2D pose representation networks.

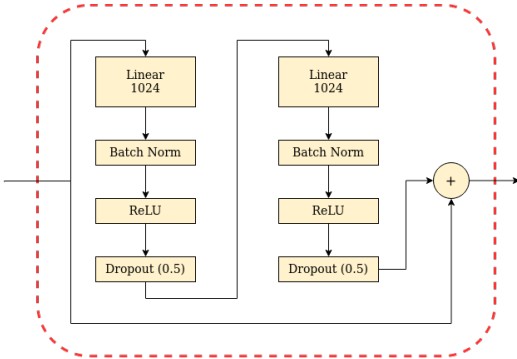

Figure 3: The residual block used within all of our 2D representation networks inspired by Martinez et al. (2017)

### A.1    FULL 2D POSE NETWORK

In Figure 4 you can see the full architecture used for our full 2D pose representation network. As we have only one network, we used one optimisation function to update the network during training. This can be seen in equation 8, where $w_1$, $w_2$ and $w_3$ are 1, 10 and 4 respectively.

$$\mathcal{L} = w_1\mathcal{L}_{adv} + w_2\mathcal{L}_{2D} + w_3\mathcal{L}_{90°} \tag{8}$$

## A.2 SPLIT-COMBINE LEG AND TORSO NETWORK

In Figure 5 you can see the full architecture used for our split-combine leg and torso network. In this scenario the full 2D pose was initially split into the leg and torso key points. The leg key points consisted of the ankles, knees and hips. The torso key points consisted of the wrists, elbows, shoulders, spine, neck, head, and crown key point. These were passed through two separate networks containing two residual blocks and ending with a linear layer which contained the local feature feature relationships between the key points within the leg and torso key points respectively. The number of neurons in the output layers relate to how many key points each network before it was provided to predict for. In other words the torso network provided features relating to 10 out of the 16 3D ordinates of the entire pose (where $\frac{10}{16} \times 1024 = 640$). Therefore the amount of neurons containing features we carried forward during concatenation was higher from this network to reflect this. Once the outputs were concatenated they were passed through an additional two residual blocks ending with a TanH function which predicted the all 3D ordinates of the pose. Similar to the full 2D pose representation, we have one network which predicts all 3D ordinates. Therefore we used one optimisation function and weights as Equation 8.

## A.3 INDEPENDENT LEG AND TORSO NETWORK

In Figure 6 you can see the full architecture used for our independent leg and torso network. In this representation, similar to our split-combine network, we split up a 2D pose into its respective leg and torso key points. These are then passed to two completely independent networks which predict the 3D ordinates solely for the key points they are provided. Unlike the split-combine network each network within this process is trained with its own optimisation function which contain the loss representative of the keypoints it predicts for (see section 3.3) Therefore the optimisation functions and their respective weights for this approach was as follows:

$$
\begin{aligned}
\mathcal{L}_{\text{legs}} &= w_1 \mathcal{L}_{adv} + w_2 \mathcal{L}_{\text{legs } 2D} + w_3 \mathcal{L}_{\text{legs } 90°} \\
\mathcal{L}_{\text{torso}} &= w_4 \mathcal{L}_{adv} + w_2 \mathcal{L}_{\text{torso } 2D} + w_3 \mathcal{L}_{\text{torso } 90°}
\end{aligned}
\tag{9}
$$

where $w_1$, $w_2$, $w_3$ and $w_4$ are $\frac{6}{16}$, 10, 3 and $\frac{10}{16}$ respectively. The discrepancy between $w_1$ and $w_4$ stems from the fact that the adversarial loss provide to each model was the same (see section 3.5) however as the torso network predicted $\frac{10}{16}$ of the 3D ordinates for the full pose, its adversarial loss is higher to reflect that any incorrect pose is more likely to be because of this network.

## A.4 SPLIT-COMBINE 5 LIMBS NETWORK

In Figure 7 you can see the architecture of our split-combine 5 limbs network. In this scenario the full 2D pose was initially split into 5 limbs comprising of the left arm, right arm, left leg, right leg and torso. Each limb contained 3 key point coordinates aside from the torso which contained 4. We first passed through each limb through its own residual block which learnt local group features. The outputs of these network were passed through a linear layer. The number of neurons in each output linear layer corresponds to how many key points that residual block had as input, so for each limb with 3 key points it has 192 neurons ($\frac{3}{16} \times 1024$) and for the torso as this has 4 key points it is 256. These features are then concatenated and passed through a final residual block which learns how to combine these features, and finally all 3D ordinates are predicted via a TanH function. As we have one network which predicts the entire 3D ordinates of a pose, therefore we use the same optimisation function and weights as in Equation 8.

## A.5 INDEPENDENT 5 LIMBS NETWORK

In Figure 8 you can see the architecture of our independent 5 limbs network. In this scenario the full 2D pose was initially split into 5 limbs as stated previously. Each limb was passed through its own residual block network which is followed by a TanH function which outputted the 3D ordinates belonging to that specific limb. These outputs were concatenated and combined with the input 2D pose to obtain our full 3D pose. As we are training 5 networks independently of one another they each receive their own specific optimisation function which contains the losses relative to the key

points that they predict for (see section 3.3). For this network the 5 optimisation functions and weights were as follows:

$$
\begin{aligned}
\mathcal{L}_{\text{left leg}} &= w_1 \mathcal{L}_{adv} + w_2 \mathcal{L}_{\text{left leg } 2D} + w_3 \mathcal{L}_{\text{left leg } 90°} \\
\mathcal{L}_{\text{right leg}} &= w_1 \mathcal{L}_{adv} + w_2 \mathcal{L}_{\text{right leg } 2D} + w_3 \mathcal{L}_{\text{right leg } 90°} \\
\mathcal{L}_{\text{left arm}} &= w_1 \mathcal{L}_{adv} + w_2 \mathcal{L}_{\text{left arm } 2D} + w_3 \mathcal{L}_{\text{left arm } 90°} \\
\mathcal{L}_{\text{right arm}} &= w_1 \mathcal{L}_{adv} + w_2 \mathcal{L}_{\text{right arm } 2D} + w_3 \mathcal{L}_{\text{right arm } 90°} \\
\mathcal{L}_{\text{torso}} &= w_4 \mathcal{L}_{adv} + w_2 \mathcal{L}_{\text{torso } 2D} + w_3 \mathcal{L}_{\text{torso } 90°}
\end{aligned}
\tag{10}
$$

where $w_1$, $w_2$, $w_3$ and $w_4$ are $\frac{3}{16}$, 10, 3 and $\frac{4}{16}$ respectively. The discrepancy between weights $w_1$ and $w_4$ is due to the fact that the all limb networks predict 3 ordinates but the torso network predicts 4.

## B    IMPROVING THE SELF-CONSISTENCY CYCLE

We found during our study that we are able to achieve a better quantitative error by improving self-consistency even if this leads to a pose that is easy to discriminate against. This can be seen clearly in Table 3 where we see a noticeable decrease in MPJPE between the results in Drover et al. (2019) and our re-creation with the additional consistency constraints mentioned within our work.

Table 3: The results of Drover et al. (2019) and our recreation with our additional consistency constraints and changing the weight values in the optimisation function to increase the importance of self-consistency.

| Model | Avg. |
|---|---|
| Drover *et al.*Drover et al. (2019) | 38.2 |
| Ours (Improved Consistency) | **34.9** |

Because of this, we sought to replace the random rotation self-consistency cycle with something more efficient. This was due to a random rotation lending itself to long training times, where the longer a model is trained the more random rotations it will see and therefore the more consistent it will become. However, this could be a problem as highlighted in Figure 2 where a longer training period may lead to GAN instability. By contrast our $90°$ consistency constraints allows for 3 specified angles of consistency to be learned per training iteration, while also being more computational efficient then randomly rotation a 3D object and reprojecting it. These by themselves however aren't sufficient to learn self-consistency as the model only learns 3 specific angles during training and in the wild many more viewpoints exist. We therefore sought an optimisation formula similar to our $90°$ consistency that would satisfy all possible viewpoints around the $y$ axis. First let us determine the end position of the pose $(\mathbf{x}, \mathbf{y}, \hat{\mathbf{z}}_i)$ after a random 3D rotation $\mathbf{R}_\theta$ along the $y$ axis:

$$
\begin{aligned}
\mathbf{R}_\theta &= \begin{bmatrix} cos(\theta) & 0 & -sin(\theta) \\ 0 & 1 & 0 \\ sin(\theta) & 0 & cos(\theta) \end{bmatrix} \\
\mathbf{R}_\theta \begin{bmatrix} \mathbf{x} \\ \mathbf{y} \\ \hat{\mathbf{z}} \end{bmatrix} &= \begin{bmatrix} \mathbf{x}cos(\theta) - \hat{\mathbf{z}}sin(\theta) \\ \mathbf{y} \\ \mathbf{x}sin(\theta) + \hat{\mathbf{z}}cos(\theta) \end{bmatrix}
\end{aligned}
\tag{11}
$$

we substitute these new positions within our network function (Equation 1)

$$
\mathbf{x}sin(\theta) + \hat{\mathbf{z}}cos(\theta) = G(\mathbf{x}cos(\theta) - \hat{\mathbf{z}}sin(\theta), \mathbf{y}; \mathbf{w})
\tag{12}
$$

for $\theta << 1$:

$$
\mathbf{x}\theta + \hat{\mathbf{z}} = G(\mathbf{x} - \mathbf{z}\theta, \mathbf{y}; \mathbf{w})
\tag{13}
$$

perform Taylor series expansion while ignoring terms of power 2 and above for the small angle $\theta$:

$$\mathbf{x}\theta + \hat{\mathbf{z}} = G(\mathbf{x}, \mathbf{y}; \mathbf{w}) - \hat{\mathbf{z}}\theta\frac{\partial}{\partial\mathbf{x}}G(\mathbf{x}, \mathbf{y}; \mathbf{w}) \tag{14}$$

cancel $\mathbf{z}$ with $G(\mathbf{x}, \mathbf{y}; \mathbf{w})$ and remove $\theta$:

$$\mathbf{x} = -\hat{\mathbf{z}}\frac{\partial}{\partial\mathbf{x}}G(\mathbf{x}, \mathbf{y}; \mathbf{w}) \tag{15}$$

this leaves us with our final consistency constraint for all angles:

$$\mathbf{x} + \hat{\mathbf{z}}\frac{\partial}{\partial\mathbf{x}}G(\mathbf{x}, \mathbf{y}; \mathbf{w}) = 0 \tag{16}$$

Although theoretically simple, in practice the above is difficult to implement and leads to lackluster performance. This is due to two factors; firstly the derivative component is a Jacobian matrix, which to calculate numerically within current deep learning languages is computationally inefficient, requiring over 100 minutes to train one epoch on an RTX-8000 GPU. Secondly as we are finding the derivative with respect to the inputs, to maintain gradient independence all batch-norm layers have to be removed from our model as these normalises across the batch dimension. This has the effect of lowering the rate at which our model learns and decreasing its stability while training Santurkar et al. (2018). We are currently investigating if layer norm Ba et al. (2016) would be a suitable replacement for batch norm as this would retain gradient independence and allow the implementation of the above within future work.

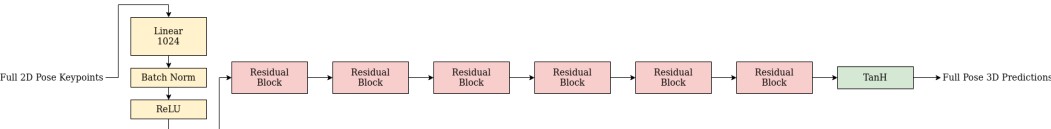

Figure 4: The architecture of our full pose lifting network

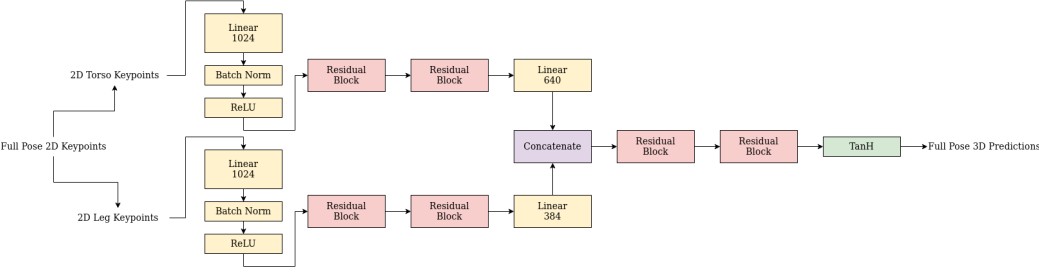

Figure 5: The architecture of our split-combine Leg and Torso Network

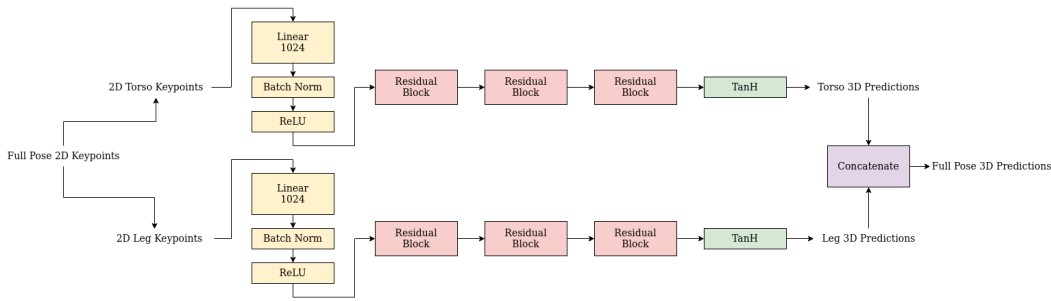

Figure 6: The architecture of our independent Leg and Torso Network

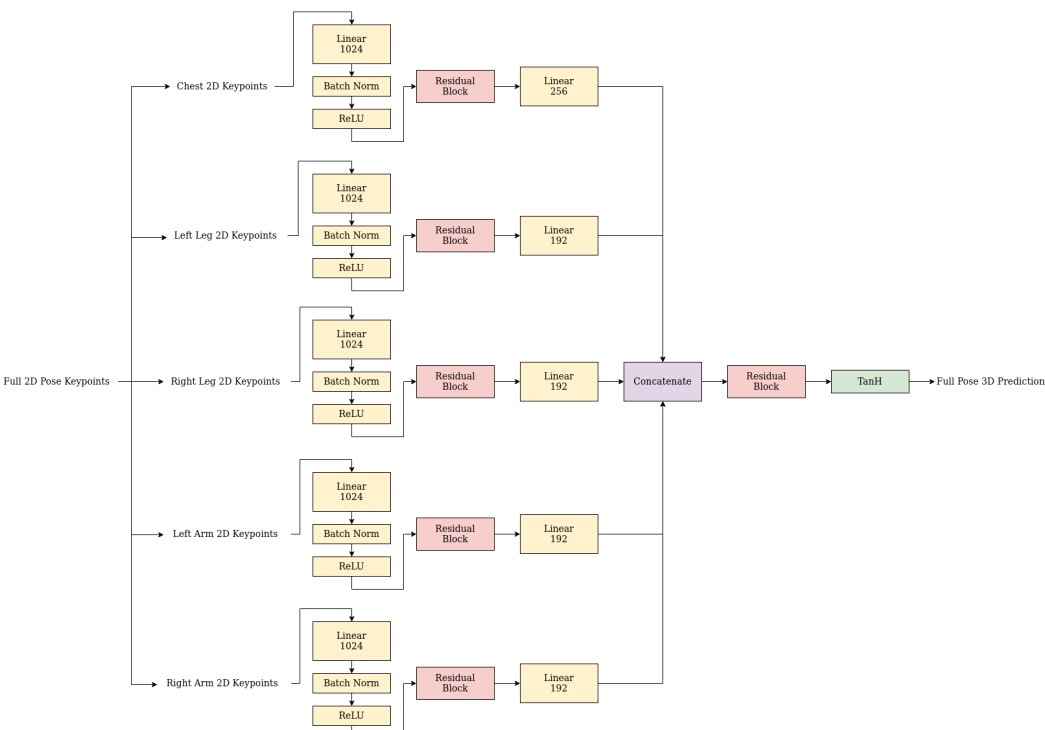

Figure 7: The architecture of our split-combine 5 limbs Network

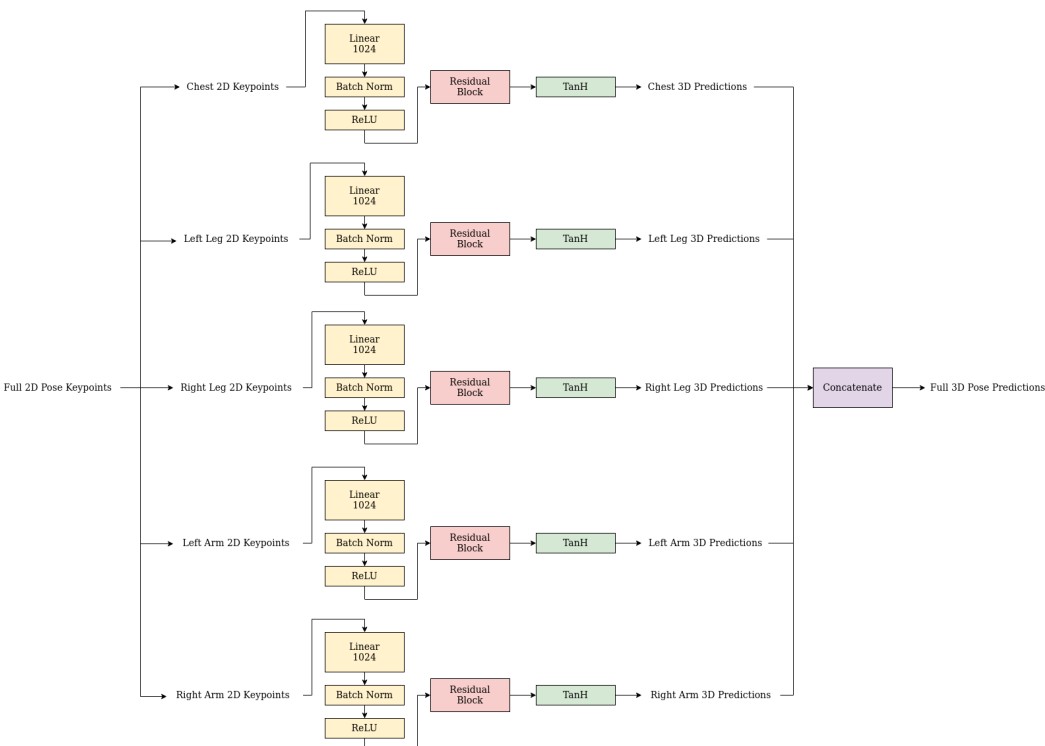

Figure 8: The architecture of our 5 independent limbs Network

