# OpenReview forum: "Optimising 2D Pose Representation: Improving Accuracy, Stability and Generalisability inUnsupervised 2D-3D Human Pose Estimation"
_ICLR.cc/2023/Conference — Submitted to ICLR 2023_

### Official Review · Reviewer_QAQe · 2022-10-23

**Confidence:** 4
**Clarity, Quality, Novelty And Reproducibility:** The proposed idea is clear and easy t…
**Correctness:** 4
**Technical Novelty And Significance:** 2
**Empirical Novelty And Significance:** 2
**Recommendation:** 3

**Strength And Weaknesses:**

Strength:
- Experimental results are promising, which verify the effectiveness of the proposed split-and-recombine strategy.

Weaknesses:
- Novelty is relatively limited. Divide-and-conquer is a common approach in computer vision. It is interesting to observe that joint grouping can improve the accuracy significantly. But I expect more in this work. For example, the authors could investigate novel methods to decently encourage the interaction between joins belonging to one group and suppress the interaction between joins belonging to different groups, in order to improve the efficiency.
- In my understanding, the proposed method should be applicable for all 2D-to-3D pose lifting methods. Why do the authors only focus on the unsupervised one? What about the experimental results on supervised and weakly-supervised methods?


**Summary Of The Paper:**

This paper proposes a method for the problem of 2D to 3D pose lifting. The authors propose to divide the all joints of a pose into several groups, and then use separate networks to lift each group of joints independently. A recombining network then takes as input the features of lifting networks and estimates the full 3D pose. The above grouping strategy is applied to the unsupervised 2D-to-3D pose lifting task. Experiments are implemented on two datasets, and the results show that the split-and-recombine method significantly outperforms the full pose lifting method using the same backbone model, and also performs better than unsupervised SOTAs. When trained on HUMAN3.6M and tested on MPI-INF-3DHP, the proposed method outperforms most of supervised and weakly-supervised competitors.

**Summary Of The Review:**

The proposed divide-and-conquer strategy achieves promising performance, but I expect the authors obtain more research achievement based on this observation.

---

### Official Review · Reviewer_2qG2 · 2022-10-24

**Confidence:** 1
**Clarity, Quality, Novelty And Reproducibility:** N/A
**Correctness:** 4
**Technical Novelty And Significance:** 4
**Empirical Novelty And Significance:** Not applicable
**Recommendation:** 8

**Strength And Weaknesses:**

N/A

**Summary Of The Paper:**

N/A

**Summary Of The Review:**

N/A

---

### Official Review · Reviewer_rPD4 · 2022-10-24

**Confidence:** 4
**Clarity, Quality, Novelty And Reproducibility:** The explanations are clear enough for…
**Correctness:** 3
**Technical Novelty And Significance:** 3
**Empirical Novelty And Significance:** 2
**Recommendation:** 5

**Strength And Weaknesses:**

Reprojection loss is novel and well motivated to refine the poses in the aspect of different camera viewpoints.

Minor errata: “This intrinsic approach allowed joint relationships to be learnt0 over
time as each key point was added.” in page 3.

Table3 in page 14 does not show how much it is improved in the aspect of the depth. I recommend authors to add this.

I’m curious how much the variations on leg poses affect the depth values of the arms. I recommend authors to report any additional results that can explain this.

There is lack of qualitative results and ablative studies. I recommend adding these.


**Summary Of The Paper:**

This proposes the unsupervised 2D-to-3D lifting approach by exprimenting with 5 network structures: 1) Popularly used 2D-to-3D lifting model, 2) extracting features from body and legs and estimate 3D full pose by concatenating 2 features, 3) Separately estimating poses for body and legs, 4) Separately estimate 5 part features (2 arms, 2 legs and body) and estimate 3D full pose using them, 5) Estimate 3D pose for 5 different parts. Further the consistency loss is proposed. Experimental results demonstrate the superiority of the proposed method.


**Summary Of The Review:**

 I think this paper is proposing some novel method; while the qualitative results and ablative studies are lacking. At this moment, I am currently at marginally below threshold for this paper; while I could change mine to the higher score if authors could rebut my points.

---

> ### Author Response · Authors · 2022-11-15
> **Response to Reviewer**
>
> We appreciate the reviewer providing useful feedback and acknowledge interest in further details regarding our findings. Firstly thank you for finding grammatical errors, they are easily missed and they are updated.
>
> With regards to Table3 not showing how much is improved in terms of depth, we do not predict absolute depth within our work but simply the depth off-set relative to the root coordinate. Because of this, our results show the error between the ground truth and predicted key-point location once rigid alignment has been conducted. Therefore were unable to say exactly how much the error is improved in the aspect of absolute depth simply that the relative depth prediction has decreased. We hope that this has provided clarification but if it has not please let us know and we would happily discuss it further.
>
> "I’m curious how much the variations on leg poses affect the depth values of the arms. I recommend authors to report any additional results that can explain this."
>
> We are actually currently exploring this avenue within future work where we are using our model in scenarios with occlusion. Our early results surprisingly show that legs aren't actually that useful when trying to predict an arm's 3D location when it is occluded. However, arms are very useful when trying to predict an occluded legs 3D location. Perhaps this is due to the arms of someone shifting to counterweight the head which dictates where their legs should be? Though this could also be dataset dependent as were noticing the right arm is more useful than the left arm due to our subjects being right-handed. Again, this is all currently under investigation but it is definitely a topic we are delving into.
>
>
> There is lack of qualitative results and ablative studies. I recommend adding these.
>
> As mentioned in our comment to reviewer 1 sadly we are unable to provide in-depth ablation studies due to the time it would take to complete but do agree that this weakens our paper in its current format. With regards to qualitative results, we are currently in the process of creating a GitHub page of our work where we would be including multiple qualitative images and videos of our model in different scenarios.
>
> Again we thank the reviewer for their time and appreciate their feedback on our work.

---

### Official Review · Reviewer_cAex · 2022-10-31

**Confidence:** 4
**Correctness:** 3
**Technical Novelty And Significance:** 2
**Empirical Novelty And Significance:** 3
**Recommendation:** 5

**Clarity, Quality, Novelty And Reproducibility:**

The paper is pretty well-written and well-focused on testing a single hypothesis. Where this falls a little short is with explaining/interpreting Figure 1 and Section 6 is not needed in my opinion (see comments above). The paper also provides code and the necessary parameters to reproduce the approach are mentioned in the paper.

**Strength And Weaknesses:**

Strengths:
* The paper designs a limited, small-scale study to evaluate a single design aspect of human pose regression networks in a controlled way. As such, the paper can vary factors individually and study the outcomes on reprojection error.
* The paper is overall well-written and easy to understand. I have some points regarding improving readability/clarity below.
* The paper studies the problem on two different datasets. The Human3.6M dataset is used to study the accuracy of the model on the training/testing dataset. The second dataset, MPI-INF-3DHP, is used to study generalization.

Weaknesses:
* The paper mentioned that their proposed input modality (split torso/leg groups) can be used as drop-in replacements to improve existing approaches that use full-body key points as input. However, there is no experiment in this paper that shows this to be true. In my opinion, this is an over-generalization and should be validated. As the paper tries to encourage the use of a different input form through a controlled study,  I think the paper could be made much stronger by showing evidence that the proposed method improves performance for one of the multiple state-of-the-art models. In my opinion, this should be rather easy to validate and will add a lot to the paper.
* The paper mentions multiple times that the proposed 3D pose representation is "optimal", implying that there can be no other representation that is better. As there is no evidence to support this claim, please remove it. Instead, the paper could refer to it as: "improved 2D representation".
* Figure 1: It is unclear to me what this figure shows. What are the colors representing? And how can the paper claim to see a clear correlation between the two selected joints? Explain more, please.
* Section 6. The paper mentions that their paper is truly unsupervised and therefore it is not entirely clear when to stop training. However, the method does NEED ground truth 2D poses to train the GAN and therefore is not fully unsupervised and this argumentation is not well supported. If 2D ground truth poses are not available, how would you train your discriminator? If this is not possible, why not just use a portion of the data for validation? In my opinion, this space could be better used by showing that the proposed approach benefits existing methods too.

Minor concerns:
* Section 3.1: the paper mentions ordinates instead of coordinates. Maybe a search/replace error?
* Table 2 caption: uses transfer learning during from...

**Summary Of The Paper:**

The paper proposes a study investigating the influence of different representations of 2D key points to regress 3D depth in an unsupervised fashion. The hypothesis studied is that when the whole 2D skeleton is input into a lifting network, the network extracts undesirable correlations between key points, which hurt performance. The most successful proposed model splits key points into two groups (torso and legs) and regresses depth individually for each group.

**Summary Of The Review:**

Overall, the paper empirically tests a hypothesis in a pretty rigorous fashion. Some of the statements have to be modified/weakened and the authors should consider testing their method with different state-of-the-art models to support their claims. I also recommend rethinking the paper layout a bit and putting the main focus on the main contributions.

---

> ### Author Response · Authors · 2022-11-15
> **Response to Reviewer**
>
> We would like to thank the reviewer for their useful feedback and for taking the time to evaluate our work. Below we provide comments addressing any concerns.
>
>
>
> 1.) The paper mentioned that their proposed input modality (split torso/leg groups) can be used as drop-in replacements to improve existing approaches that use full-body key points as input. However, there is no experiment in this paper that shows this to be true...
>
> We definitely agree. The main message we want the reader to take away from this paper is the potential to use our findings into current work, and showing the improvement would have been very useful. Due to the difficulty of re-creating prior work, either due to no code or lack of detail in the paper (as we mentioned in section 6 due to training length), we are currently unable to provide any true re-creations. However, we would like to highlight that our work uses the same residual block architecture used in several other studies [1][2][3] which all utilize the same adversarial training approach. Given the only thing that changes between our study and their own are additional constraints on a similar objective function, we would expect our findings to improve their work also.
>
>
>
> 2.) The paper mentions multiple times that the proposed 3D pose representation is "optimal"...
>
> We agree with the reviewer that saying 'optimal' is not suitable as further research could find a better pose representation. This has been updated to 'most improved representation' in our rebuttal version.
>
>
>
>
> 3.) Figure 1: It is unclear to me what this figure shows. What are the colors representing? And how can the paper claim to see a clear correlation between the two selected joints? Explain more, please
>
> We agree with the reviewer that the explanation of Figure 1 can be improved and we have updated it in the rebuttal version. The x-axis of the Figure shows how much the 2D key-point we have decided to change has been scaled by (-5 - 5%). The y-axis shows the change in the prediction of our model due to this 2D key-point scaling. As for the correlation we mentioned, the top-right figure shows how our solo models right ankle 3D ordinate prediction changes when the left elbow 2D coordinate is scaled. As the 3D ordinate prediction is altered by a change in the left elbows 2D key-point location we can say that these are correlated as a change in one has led to a change in the other. This correlation is removed in the bottom-right figure (torso legs split model) which we believe is why we have obtained better results. The colours by themselves do not represent anything we have simply included them to easily discriminate between the density distributions and can be removed.
>
>
>
>
>
> 4. ) Section 6. The paper mentions that their paper is truly unsupervised and therefore it is not entirely clear when to stop training. However, the method does NEED ground truth 2D poses to train the GAN and therefore is not fully unsupervised and this argumentation is not well supported. If 2D ground truth poses are not available, how would you train your discriminator...?
>
> We understand the confusion with stating our approach as unsupervised when we use ground truth 2D poses to train. To begin, we would like to clarify that we do not need ground truth 2D poses to train our model,  only believable 2D poses. We could have easily trained our model on 2D poses obtained from a 2D key-point detector as mentioned in the related work. Additionally, we highlight that prior work [2][4] also state their model as unsupervised but similarly use ground truth 2D key-points during training and evaluation as it allows for a fair comparison due to the discrepancy in performance when using a 2D key-point detector. By unsupervised what we mean is trying to regress the 3D depth offset with only 2D data as 2D data. We have added additional clarification about this in the introduction stating "by unsupervised we mean regressing 3D ordinate positions while only using 2D data which is easier to obtain..."
>
> [1] Chen, Ching-Hang et al. “Unsupervised 3D Pose Estimation With Geometric Self-Supervision.” 2019 IEEE/CVF Conference on Computer Vision and Pattern Recognition (CVPR) (2019): 5707-5717.
> [2] Drover, D., M. V, R., Chen, CH., Agrawal, A., Tyagi, A., Huynh, C.P. (2019). Can 3D Pose Be Learned from 2D Projections Alone?. In: Leal-Taixé, L., Roth, S. (eds) Computer Vision – ECCV 2018 Workshops. ECCV 2018. Lecture Notes in Computer Science(), vol 11132. Springer, Cham. https://doi.org/10.1007/978-3-030-11018-5_7
> [3] Wandt, Bastian & Rosenhahn, Bodo. (2019). RepNet: Weakly Supervised Training of an Adversarial Reprojection Network for 3D Human Pose Estimation. 7774-7783. 10.1109/CVPR.2019.00797.
> [4] Z. Yu, B. Ni, J. Xu, J. Wang, C. Zhao and W. Zhang, "Towards Alleviating the Modeling Ambiguity of Unsupervised Monocular 3D Human Pose Estimation," 2021 IEEE/CVF International Conference on Computer Vision (ICCV), 2021, pp. 8631-8631, doi: 10.1109/ICCV48922.2021.00853.

---

### Official Review · Reviewer_p3BW · 2022-11-01

**Confidence:** 5
**Clarity, Quality, Novelty And Reproducibility:** Good
**Correctness:** 4
**Technical Novelty And Significance:** 3
**Empirical Novelty And Significance:** Not applicable
**Recommendation:** 5

**Strength And Weaknesses:**

Strength:
1. The author has validated the effectiveness of reducing unintuitive key point correlations. The model using an independent leg and torso representation performs better than using the full points. The proposed idea is very simple but effective.
2. A new matrix, 90-degree consistency, is proposed. As far as I know, this matrix is novel.
3. Detailed explanation of the model and experiment parameters, good for reproducing.

Weakness
1. My main concern is the performance compared to the state-of-the-art. In the results on the MPI-INF-3DHP in Table  2, the best one of the proposed models (leg and torso) has a PCK3D of 78.5, still less than the 82.2 of the model by Yu et al.
2. Can the author provide an ablation study to show the effectiveness of each of the loss components, saying adversarial, reprojection consistency, and 90-degree consistency losses? It would be interesting to see the effectiveness of each one and the interaction between them.


**Summary Of The Paper:**

This paper conducted investigation on how different 2D pose representations affects the unsupervised adversarial 2D-3D lifting process. Experiments are conducted on both Human3.6M dataset and MPI-INF-3DHP dataset. The results show that reducing the unintuitive key point correlations can reduce the average error. In addition, the paper explores the factors that affects the convergence and adversarial training stability.

**Summary Of The Review:**

The experiments are conducted on two datasets and are valuable as a case study. However, the improvement over the baselines looks marginal. Also, the effectiveness of each of the loss components is not very clear. I believe that addressing my points of weakness to make them more clear would largely benefit the paper.

---

> ### Author Response · Authors · 2022-11-15
> **Response to Reviewer**
>
> We thank the reviewer for taking the time to read our work and would like to address their comments below.
>
> 1.) My main concern is the performance compared to the state-of-the-art. In the results on the MPI-INF-3DHP in Table 2, the best one of the proposed models (leg and torso) has a PCK3D of 78.5, still less than the 82.2 of the model by Yu et al.
>
> We acknowledge that compared to some state-of-the-art models our results are inferior however, we would like to highlight the comparison between Yu et al. and ourselves is not entirely fair as they are a temporal model utilizing a sequence of poses as input with a temporal motion consistency constraint whereas we use one pose as input with no temporal information. Additionally, they state within their work "that adding temporal motion consistency can boost the performance by about 6%" so we believe that if ours and their findings were combined the results would also improve.
>
>
> 2.) Can the author provide an ablation study to show the effectiveness of each of the loss components, saying adversarial, projection, consistency and 90-degree consistency losses? It would be interesting to see the effectiveness of each one and the interaction between them.
>
> We agree with the reviewer that a lack of an ablation study showing the impact of each loss component definitely makes the findings and takeaway of our paper weaker. As we are presenting the findings of multiple different models we would want to show the ablations on each. Sadly this would take time to complete but if possible we could provide it in the supplementary material in a camera-ready paper. In the meantime we would like to highlight the work by Chen et al. "Unsupervised 3D Pose Estimation with Geometric Self-Supervision" where they provide a detailed ablation study about the adversarial and re-projection losses, our work is similar in architecture so we would also expect the results to show similar improvements.

---

> > ### Comment · Reviewer_p3BW · 2022-11-28
> > **Response to Paper2537 Authors**
> >
> > Thank you Paper2537 Authors for the detailed response. Regarding the replies:
> >
> > 1. The authors mention that the SoTAs are using a sequence of poses as input while they only use one pose. Can the authors update the table by grouping the methods by "temporal" and "non-temporal" models, so the readers can compare your model with non-temporal ones? Also, given that you are not using temporal information, I would be curious about the time complexity and memory required for your model.
> >
> > 2. I understand adding additional experiments would take time to complete.
> >
> > I am convinced that the paper would have better quality if given more time. However, I still think more needs to be done to explain the impact of each loss component. Given the current version, the paper is not ready for publication. Overall, I vote for borderline.

---

### Decision · Program_Chairs · 2023-01-20

**Decision:**

Reject

**Justification For Why Not Higher Score:**

Lots of important experiments are missing.

**Justification For Why Not Lower Score:**

NA

**Metareview: Summary, Strengths And Weaknesses:**

This paper focuses on unsupervised 2D to 3D pose lifting.  Some novel designs are proposed in this paper, which include the 90-degree consistency (as pointed out by reviewer p3BW), the reprojection loss (as pointed out by rPD4). However, some important qualitative results and ablation studies (e.g., the efficacy of each module, such as the reprojection loss) are lacking. Thus, four of the five reviewers are negative on this paper. Note that Reviewer 2qG2 gave the score 8, but he/she contacted the AC and mentioned that he/she is not in this area, so not able to provide comments on this paper. Thus Reviewer 2qG2's recommendation is not considered in the decision. Considering that the work still needs to be significantly improved especially on the experiment part, AC recommends rejection for this paper, but highly encourages the authors to improve the paper following reviewers' comments for resubmission to another venue.